# The Missing Tailed Phages: Prediction of Small Capsid Candidates

**DOI:** 10.3390/microorganisms8121944

**Published:** 2020-12-08

**Authors:** Antoni Luque, Sean Benler, Diana Y. Lee, Colin Brown, Simon White

**Affiliations:** 1Viral Information Institute, San Diego State University, San Diego, CA 92182, USA; nuevecuervos@gmail.com (D.Y.L.); colintravisbrown@gmail.com (C.B.); 2Computational Science Research Center, San Diego State University, San Diego, CA 92182, USA; 3Department of Mathematics and Statistics, San Diego State University, San Diego, CA 92182, USA; 4National Center for Biotechnology Information (NCBI), Bethesda, MD 20894, USA; sbenler@gmail.com; 5Department of Physics, San Diego State University, San Diego, CA 92182, USA; 6Department of Molecular and Cell Biology, University of Connecticut, Storrs, CT 06269, USA; simon.white@uconn.edu

**Keywords:** bacteriophage, tailed phages, icosahedral capsids, capsid modeling, statistical learning, isolated genomes, metagenome-assembled genomes

## Abstract

Tailed phages are the most abundant and diverse group of viruses on the planet. Yet, the smallest tailed phages display relatively complex capsids and large genomes compared to other viruses. The lack of tailed phages forming the common icosahedral capsid architectures T = 1 and T = 3 is puzzling. Here, we extracted geometrical features from high-resolution tailed phage capsid reconstructions and built a statistical model based on physical principles to predict the capsid diameter and genome length of the missing small-tailed phage capsids. We applied the model to 3348 isolated tailed phage genomes and 1496 gut metagenome-assembled tailed phage genomes. Four isolated tailed phages were predicted to form T = 3 icosahedral capsids, and twenty-one metagenome-assembled tailed phages were predicted to form T < 3 capsids. The smallest capsid predicted was a T = 4/3 ≈ 1.33 architecture. No tailed phages were predicted to form the smallest icosahedral architecture, T = 1. We discuss the feasibility of the missing T = 1 tailed phage capsids and the implications of isolating and characterizing small-tailed phages for viral evolution and phage therapy.

## 1. Introduction

Tailed phages are viruses that infect bacteria and are the most abundant biological entity on Earth [1]. They are responsible for the regulation of biogeochemical processes at a planetary scale [2,3], the control of microbial populations [4,5], and the mobility of genes across hosts and ecosystems [6,7]. The broad functionality of tailed phages is facilitated by their vast reservoir of genes, resulting in a large range of genome lengths, from 10 to 500 kilobase pairs (kbp) [8,9]. Tailed phages can accommodate these genomes because the proteins building their protective capsids display a high degree of plasticity, allowing for a broad range of different size capsids, from about 30 to 160 nm in diameter [10]. The majority of tailed phages, about 80–90%, form capsids with icosahedral symmetry [11], while the remaining 10–20% form elongated capsids with icosahedral caps [12,13]. The formation of icosahedral capsids is not unique to tailed phages—it is the most common viral capsid architecture in the virosphere [14]. However, despite the abundance and diversity of tailed phages, even the smallest tailed phages form far more complex capsids and store far larger genomes than most icosahedral viral families [15]. Furthermore, the major capsid proteins of tailed phages adopt the HK97-fold, which is also found in the building blocks of bacterial and archeal enzymatic cellular nanocompartments called encapsulins [16,17]. Intriguingly, encapsulins can form the smallest icosahedral protein shells that are absent among tailed phages (Figure 1).

Icosahedral capsids are characterized by the triangulation number T, which determines the number of quasi-equivalent proteins (complexity) and the total number of proteins forming the capsid [18,19]. As illustrated in Figure 2, icosahedral capsids can organize their proteins in four different lattices, accommodating different stoichiometries of major and minor capsid proteins [19]. The generalized T-number is T_i_(h,k) = α_i_ T_0_(h,k), where h and k are the steps in the hexagonal sublattice joining two consecutive vertices in the icosahedral capsid. T_0_ is the classic T-number associated with the hexagonal lattice and is given by the equation T_0_(h,k) = h^2^ + hk + k^2^. The subindex i takes the values h, t, s, and r, associated, respectively, with the hexagonal, trihexagonal, snub hexagonal, and rhombitrihexagonal icosahedral lattices. The number of major capsid proteins in a capsid is 60T_0_(h,k). The hexagonal lattice case only contains major capsid proteins, that is, T_h_(h,k) = T_0_(h,k), with α_h_ = 1. If the size of the major capsid protein is conserved across lattices, the other three lattices must include minor capsid proteins occupying the secondary polygons (triangles and squares). This increases the surface of the capsid by a factor α_t_ = 4/3 ≈ 1.33 (trihexagonal), α_s_ = 7/3 ≈ 2.33 (snub hexagonal), and α_r_ = 4/3 + 2/$\sqrt{3}$ ≈ 2.49 (rhombitrihexagonal). When combining all lattices, the first four elements of the generalized T-number are T = 1, 1.33, 2.33, and 2.49, containing 60 major capsid proteins each. The fifth element of the series is T = 3 and contains 180 major capsid proteins. Tailed phages have been observed to form capsids adopting the hexagonal and trihexagonal lattices [19,20], but no tailed phages have been observed to form T ≤ 3 capsids (Figure 1). The smallest characterized tailed phage structure corresponds to the *Bacillus* phage phi29, which adopts an elongated structure with icosahedral T = 3 caps and a Q = 5 body [12,21,22], and *Streptococcus* phage C1, which adopts a T = 4 icosahedral capsid [10,23].

There are several interrelated scientific observations that make the lack of small icosahedral capsids striking among tailed phages. First, T = 1 and T = 3 icosahedral capsid architectures are optimal thermodynamic configurations for protein shells and are the most kinetically favorable icosahedral capsids [13,24,25,26,27]—they are the most common architectures imaged among viruses [15]. Second, tailed phages belong to the *Duplodnaviria* viral realm, which also includes archaeal and eukaryotic viruses [28,29]. These viruses are characterized by building their capsids with a major capsid protein adopting the highly conserved HK97-fold, and packing their genome in the form of double-stranded DNA at quasi-crystalline densities (Figure 1). The structural elements of this realm appear to have emerged prior to the Last Universal Cellular Ancestor [30]. Modern tailed phages forming T = 4 and larger capsid shells are expected to have evolved from precursor capsids that used T = 1 and T = 3 architectures. Third, encapsulins organize the HK97-fold in T = 1, T = 3, and T = 4 architectures, showing that the HK97-fold is capable of forming small icosahedral capsids [16,17] (Figure 1). Finally, non-tailed icosahedral phages in the *Microviridae* family adopt T = 1 capsids, storing a small genome capable of executing a lytic replication strategy analogous to tailed phages [31,32]. This suggests that tailed phages should be able to store the instructions for their lytic lifestyle in T = 1 capsids.

This body of evidence indicates that T = 1 and T = 3 should exist among tailed phages. Our working hypothesis is that these small-tailed phages have not been isolated because they are relatively low in abundance in most environments sampled. Larger tailed phage capsids capable of storing longer genomes may have been favored by incorporating genes that can alter the host’s physiology, protect the bacterial host against other phages, and overcome the host’s resistance mechanisms [6,33,34,35,36]. Here, we applied modeling and bioinformatics to narrow the search for small-tailed phages, overcoming the current limitations of sampling. As a proof of concept, we analyzed the capsid architecture and genome size of twenty-three tailed phages that have had high-resolution structures of their capsid determined. We used an allometric model based on conserved structural characteristics of tailed phages and trained the model to infer the genome size and capsid diameter size of tailed phages as a function of the T-number icosahedral architecture. The predictions of the model were compared with more than 3348 isolated phage genomes and 1496 gut metagenome-assembled tailed phage genomes. Four isolated tailed phages and twenty-one metagenome-assembled tailed phages were predicted to form, respectively, T = 3 and T < 3 icosahedral capsids’ architectures, currently missing among tailed phages.

## 2. Materials and Methods

Structural features of tailed phage capsids: High-resolution tailed phage structures were obtained from twenty-three cryo-EM maps [23,37,38,39,40,41,42,43,44,45,46,47,48,49,50,51,52,53,54,55,56,57,58]. UCSF Chimera software was used to visualize the structures and measure the geometrical properties of the capsids [59]. The icosahedral tool was fitted manually to the internal and external surface of the capsids by adjusting the radius and sphericity parameters. Error measurements were assigned by comparing the outputs at one discrete step below and above the final measurement. The direct measurements were the internal radius, sphericity, surface, and volume, as well as external radius, sphericity, surface, and volume. The shell thickness was estimated by subtracting the internal radius from the external radius. The number of major capsid proteins was calculated using the icosahedral T-number associated with each capsid [10,18,19]. The internal and external major capsid protein surfaces were obtained by dividing the fraction of internal and external surfaces associated with major capsid proteins with respect to the total number of major capsid proteins. The genome density was estimated by dividing the genome size by the internal volume. The Electron Microscopy Database ID (EMDB) and measurements obtained for each tailed phage are provided in Appendix A. The correlation with the capsid diameter of all variables measured was evaluated as a function of the interior and external capsid diameters using the non-parametric Spearman coefficient correlation and a two-tailed test. The normality of the variables that were not correlated with capsid diameter was assessed using the Shapiro–Wilk test.

Statistical model: Allometric models were used to relate the genome length (G) and the capsid diameter (D) as a function of the capsid architecture, T-number. To reduce the bias associated with the oversampling of some T architectures, such as T = 7 capsids, the model was built using the average genome lengths and capsid diameters for each T-number. The models were, respectively, $G\left(T\right)=a_{G}T^{b_{G}}$ and $D\left(T\right)=a_{D}T^{b_{D}}$. Each model had two parameters: the pre-factor $a_{i}$ and the allometric exponent $b_{i}$. The subindex i took the values G for the genome length model and D for the capsid diameter model. The variables were logged on base 10 to obtain the best fit using a linear regression least-squares approach for the parameters $\log(a_{i})$ and $b_{i}$. The residuals were analyzed to evaluate the accuracy of the standard errors and confidence intervals extracted from the linear regression analysis, which relied on the standard assumptions of linearity, normality, homoscedasticity, and low leverage [60,61]. The linear statistical analysis and predictions were performed using the *lm* and *predict* functions in the statistical computing language R [62].

Three-dimensional (3D) icosahedral capsid models: Small capsid 3D model architectures with T-number T ≤ 4 were generated for hexagonal (h), trihexagonal (t), snub hexagonal (s), and rhombitrihexagonal (r) icosahedral lattices. These Archimedean lattices minimize the structural information necessary for building icosahedral capsids and are the basis for the generalized theory of icosahedral capsids [19]. The 3D geometrical models were generated computationally using the new hkcage tool in ChimeraX, developed by co-authors C. Brown and A. Luque and available in https://github.com/luquelab/hkcage.git. This new tool can display any element of the infinite series of icosahedral capsids for the four fundamental lattices and their associated dual Laves lattices.

Predicted capsid architectures among tailed phage isolates: The genomes of isolated tailed phages were obtained from the NCBI Reference Sequence Database [63]. The genome information was filtered by the viral realm (*Duplodnaviria*) and host (bacteria). The database was accessed in October 2020 and downloaded in table format. For 95% of the tailed phages, the NCBI id and genome length were extracted using a Unix shell script. The remaining 5% of tailed phages had information that was not standardized with the NCBI format and was extracted manually. The final file analyzed included the NCBI identifiers and genome lengths (Appendix A). The probability density distribution of the genome lengths was obtained using Gaussian kernels with the default bandwidth for the density function in the statistical computing language R [62]. The predicted icosahedral architecture for each tailed phage was obtained by identifying the nearest average genome length obtained in the statistical T-number model described above. A frequency analysis of the predicted T-numbers was obtained. The tailed phages associated with architectures T ≤ 4 were filtered and analyzed in depth. The phage genomes associated to larger capsids were not analyzed in detail.

Predicted structures of metagenome-assembled tailed phages: The genomes of uncultured human gut tailed phages were obtained from 5742 whole-community human gut metagenomes studied by Benler et al. in Reference [64] and available in the NCBI repository: https://ftp.ncbi.nih.gov/pub/yutinn/benler_2020/gut_phages/. Only ‘circular’ contigs with relatively long direct repeats (50–200 bp) at their termini were searched for open reading frames (ORFs). The contigs were also required to include a known phage marker profile, that is, the terminase large subunit, major capsid protein, or portal protein. The contig identifier and genome length of the selected gut metagenome-assembled tailed phages are provided in Appendix A. The analysis of the distribution of complete circular genomes and predicted T-number architectures was conducted following the same protocol described above for isolated tailed phage genomes. The tailed phage genomes associated with predicted architectures T ≤ 3 were annotated using the same procedure described in Reference [64]. The phage genomes associated to larger capsids were not analyzed in detail.

## 3. Results

### 3.1. Structural Properties of Characterized Tailed Phage Capsids

The range of structural properties obtained from the twenty-three capsids studied is summarized in Table 1. The T-number ranged from 4 to 27, containing 240 to 1620 major capsid proteins, although 5 of those major capsid proteins were replaced by portal proteins in the virion. The external capsid surface was relatively angular, with sphericity ranging from 0.14 to 0.55. The interior capsid surfaces were more spherical, with sphericity ranging from 0.25 to 0.75. The capsid sphericity was negatively correlated with capsid diameter (rho = −0.68, *p*-value < 0.001). The largest capsid had a diameter three times bigger than the smallest capsid (143 versus 49 nm). The capsid thickness ranged from 3 to 8 nm and was positively correlated with capsid diameter (rho = 0.66, *p*-value = 0.0010). The largest interior capsid volume was about 25 times larger than the smallest one (3.36 × 10^4^ to 8.22 × 10^5^ nm^3^ range). The largest genome (280 kbp) was about 16 times larger than the smallest one (17 kbp). The average genome packing density within the capsid was approximately 0.5 bp/nm^3^, ranging from 0.34 to 0.62 bp/nm^3^. The values for the interior and exterior capsid surface areas spanned 8.5-fold (5.08 × 10^3^–4.32 × 10^4^ nm^2^) and 8-fold (6.55 × 10^3^–5.19 × 10^4^ nm^2^) respectively, with the exterior capsid surface area being 15% to 43% larger than the interior area. The average surface area for each major capsid protein in the interior and exterior parts of the capsid was approximately 23 nm^2^ (19 to 26 nm^2^) and 30 nm^2^ (24 to 35 nm^2^) per major capsid protein, respectively. The genome density (rho = −0.16, *p*-value = 0.47) and major capsid exterior surface area (rho = 0.38, *p*-value = 0.082) were the only variables that did not display a significant correlation with capsid size (see Appendix A).

### 3.2. Tailed Phage Capsids: Statistical Models and Predictions

The genome lengths and capsid diameters obtained from the high-resolution phage structures were used to build a statistical model relating the icosahedral capsid architecture with these two accessible quantities. The allometric model for the average genome length, G, as a function of the architecture index, T, explained 98.5% (R^2^ = 0.985, *n* = 7) of the variance (Figure 3a). The pre-factor was $\log a_{G}=0.37\pm 0.10$ (S.E.) with $a_{G}$ in kbp units. The allometric exponent was $b_{G}=1.47\pm 0.09$. The coefficient of variation (CV = S.E./mean) for the intercept was 27%, significantly larger than the CV associated with the power exponent, 6.1%. Similarly, the allometric model for the average capsid diameter, D, as a function of the architecture index, T, explained 98.6% (R^2^ = 0.986, *n* = 7) of the variance (Figure 3b). The pre-factor was $\log a_{D}=1.38\pm 0.34$ (S.E.) with $a_{D}$ in nm units and coefficient of variation (CV) of 25%. The allometric exponent was $b_{D}=0.52\pm 0.03$ with a CV of 5.8%. The qualitative diagnostic of the residuals was similar for both models, consistent with the standard assumptions associated with the statistics of the linear regression analysis (Appendix A). The residuals were scattered around zero with a near-normal distribution and a standardized range on the order of ±1 with relatively homoscedastic variance and relatively low leverage.

The allometric exponents obtained in the statistical models were compared with theoretical scaling relationships. The capsid structural analysis revealed that the genome density was constant. This implied that the genome length (G) was proportional to the capsid volume (V). The volume of a quasi-spherical capsid depends on the third power of the diameter (D), leading to the scaling G ~ D^3^ between the genome length and capsid diameter. The T-number, by definition, is proportional to the capsid surface, and the units are implicitly related to the major capsid protein surface [19,65]. The exposed surface of the major capsid protein obtained in the structural analysis was constant. Since the capsid surface of a quasi-spherical shell depends on the square of the diameter, this leads to the scaling T ~ D^2^ relating to the T-number and the capsid diameter. The scaling relationships derived led to the allometric relationships G ~ T^3/2^ and D ~ T^1/2^. The theoretical exponent for the genome length versus capsid architecture relationship was $b_{G}^{th}=3/2=1.5$, which was within the empirical range obtained in the statistical model, $b_{G}=1.47\pm 0.09$ (Figure 3a). The theoretical exponent for the capsid diameter versus capsid architecture yielded $b_{D}^{th}=1/2=0.5$, which was also within the empirical range obtained in the statistical model $b_{D}=0.52\pm 0.03$ (Figure 3b). The agreement between the statistical model and the theoretical allometric exponents provides confidence in using the statistical model to predict the properties of capsids for T-number architectures outside the range used to train the statistical model.

The average genome length predicted for small-tailed phage architectures was 2.34 kbp (T = 1), 3.57 kbp (T ≈ 1.33), 8.14 kbp (T ≈ 2.33), 8.94 kbp (T ≈ 2.49), 11.78 kbp (T = 3), and 17.98 kbp (T = 4). The capsid architectures and 95% confidence intervals for the genomes are shown in Figure 4. The average capsid diameter predicted for these architectures was 24.20 nm (T = 1), 28.09 nm (T ≈ 1.33), 37.54 nm (T ≈ 2.33), 38.81 nm (T ≈ 2.49), 42.76 nm (T = 3), and 49.63 nm (T = 4). The 95% confidence intervals are also displayed in Figure 4. The confidence intervals for the predicted genome lengths and capsid diameters of these small capsids were relatively large and overlapped. This was a consequence of the relatively large coefficient of variation of the intercepts in statistical models due to the limited number of different T-number capsid architectures used to generate the models (*n* = 7). The 95% confidence interval ranges for the genome lengths and capsid diameters associated with each T-number (T < 40) are provided, respectively, in Appendix A.

### 3.3. Putative Small-Tailed Phage Candidates

#### 3.3.1. Predictions from Isolated Tailed Phages

The genome lengths of 3348 isolated tailed phages were investigated. The distribution of genome lengths was multimodal (Figure 5a). The most dominant peak was at 42.0 kbp. The second dominant peak was among the largest genomes with a length of 158.9 kbp. The shortest genomes displayed a minor peak at 18.3 kbp. The minimum genome length was 11.6 kbp, while the maximum was 497.5 kbp. The median genome length was 50.3 kbp, while the mean was 72.7 kbp. The associated icosahedral structures predicted ranged from T = 3 to T = 39 (Figure 5b). The median architecture was at T ≈ 7.46, and the mean architecture value was T ≈ 9.95. The three genome length peaks were associated with predicted T ~ 4-like structures (18.3 kbp peak), T ~ 7-like structures (42.0 kbp peak), and T ~ 16–19-like structures (158.9 kbp peak). Among the smallest tailed phages, seventy-seven were predicted to form T = 4 architectures, and six were initially predicted to form T = 3 architectures. The identifiers for the isolated tailed phages predicted to form T ≤ 4 capsid architectures are provided in Appendix A. The T-number predictions for the full set of isolated tailed phages are provided in Appendix A.

#### 3.3.2. Predictions from Metagenome-Assembled Tailed Phages

The genome lengths of 1496 metagenome-assembled tailed phages obtained from gut metagenomes were investigated. The distribution of genome lengths was also multimodal (Figure 5c). The most dominant peak was at 42.9 kbp, similar to the dominant peak among isolated genomes. The second dominant peak was at 98.2 kbp, which was absent in the isolated genomes. The third dominant peak was among the largest genomes with a length of 160.8 kbp, similar to the isolated genomes. The shortest genomes displayed a minor peak at 12.6 kbp, which was slightly shorter in length compared to the analogous peak among isolated genomes. The minimum genome length was 4.5 kbp, while the maximum was 294.5 kbp. The median genome length was 44.8 kbp, while the mean was 55.8 kbp. The associated icosahedral structures predicted ranged from T ≈ 1.33 to T = 27 (Figure 5d). The median architecture was at T ≈ 7.46, and the mean architecture value was T ≈ 8.34. The four genome length peaks were associated with predicted T ~ 3-like structures (12.6 kbp peak), T ~ 7-like structures (42.9 kbp peak), T ~ 12–13-like structures (98.2 kbp peak), and T ~ 16–19-like structures (160.8 kbp peak). Among the smallest tailed phages, two were predicted to form T ≈ 1.33 architectures, nine were predicted to form T ≈ 2.33 architectures, ten were predicted to form T ≈ 2.49 architectures, forty-three were predicted to form T = 3 architectures, and forty-one were predicted to form T = 4 architectures. The identifiers for the metagenome-assembled tailed phages predicted to form T ≤ 4 capsid architectures are provided in Appendix A. The T-number predictions for the full set of metagenome-assembled tailed phages are provided in Appendix A.

#### 3.3.3. Small-Tailed Phage Capsid Candidates

Six entries in the isolated tailed phage database were predicted to form T = 3 capsid architectures. This was the smallest architecture predicted for this group. However, only four out of those six were kept in the pool of predicted T = 3 phage capsids after further scrutiny. *Enterobacteria phage P4* (NC_001609 and genome length 11.6 kbp) was discarded because it is a satellite phage that relies on the infection and genes of phage P2 to produce particles, forming a larger capsid than predicted, T = 4 [66]. *Enterobacteria phage BF23* was discarded because the entry identified in NCBI (NC_042564) is a partial genome encompassing only the tRNA gene region (gene length 14.5 kbp). The complete genome was not deposited, but the phage is T5-like, encoding a 100–120 kbp genome. The remaining four were complete phage genomes and infected hosts from four different phyla (Figure 6a). *Salmonella* phage astrithr (11.6 kbp, NC_48862) was a *Podoviridae* infecting *Salmonella enterica* in the phylum *Proteobacteria. Rhodococcus* phage RRH1 (14.3 kbp, NC_016651) was a *Siphoviridae* infecting *Rhodococcus rhodochrous* in the phylum *Firmicutes*. The two more distant phages were *Lactococcus* phage bIL311 (14.5 kbp, NC_002670), a *Siphoviridae* infecting *Lactococcus lactic* in the phylum *Actinobacteria*, and *Mycoplasma* virus P1 (11.7 kbp, NC_002515), a *Podoviridae* infecting *Mycoplasma pulmonis* in the phylum *Tenericutes*.

Twenty-one metagenome-assembled tailed phages analyzed were predicted to form T < 3 capsid architectures. No candidates, however, were predicted to form the smallest icosahedral architecture, T = 1 (Figure 6b). The 3D capsid models and labels of the contigs associated with T ≈ 1.33 (two contigs), T ≈ 2.33 (nine contigs), and T ≈ 2.49 (ten contigs) are displayed in Figure 6b. The genome architectures were diverse (Appendix A). The genomes associated with T ≈ 1.33 did not encode major capsid proteins. The smallest genome containing a major capsid protein was OLNE01004159.1, which had a genome length of 7.4 kbp and a predicted architecture of T ≈ 2.33 (Figure 6c).

## 4. Discussion

Our hypothesis was that tailed phages adopting small icosahedral structures do exist, but their low abundance in the environment has precluded isolating them to be characterized in high-resolution capsid reconstruction studies. The modeling and bioinformatic approach introduced here supports this hypothesis. The allometric models trained using tailed phage with high-resolution structures predicted capsids among isolated and metagenome-assembled tailed phages that were smaller than the icosahedral tailed phages characterized to date. Among isolated tailed phages, we predicted four T = 3 icosahedral capsids. Among metagenome-assembled tailed phages, the study predicted twenty-one potential T < 3 architectures, including two T = 4/3 ≈ 1.33 icosahedral capsids.

The agreement between the theoretical and statistical scaling exponents for the genome length and the T-number rigorously quantified the expected relationships for tailed phages. Tailed phages pack their genome at a similar density, ~0.5 bp/nm^3^. This leads to a genome length scaling with the volume of the capsid. Since the T-number is proportional to the surface of the capsid and the exposed surface of the capsid protein is conserved, this led to a scaling of 3/2 (statistically, 1.47±0.09) relating the genome length and the capsid architecture expressed in terms of the T-number. The genome length can be determined experimentally through molecular biology methods while the T-number can be obtained from high-resolution microscopy methods. The derived relationship between the genome length and T-number thus provided a direct approach to estimate the capsid architectures from the abundant molecular biology data. It also reduced the uncertainty of incorporating intermediate relationships, for example, with the capsid volume. The application of such relationship to isolated and assembled tailed phage genomes provided a method to predict the existence of missing small-tailed phage capsids.

The four isolated phages predicted to form T = 3 capsid architectures infected hosts from four distantly related phyla, suggesting that small-tailed phage capsids might be prevalent across bacterial hosts. The two isolated phages with the smallest genomes adopting T = 3 capsid architectures were *Podoviridae*: *Salmonella* phage astrithr (11.6 kbp, NC_48862) and *Mycoplasma* virus P1 (11.7 kbp, NC_002515). Their genome lengths were very similar to the predicted average genome length for T = 3 tailed phage capsids, 11.8 kbp. The other two phages were *Siphoviridae* and they had genomes ~ 3 kbp longer: *Rhodococcus* phage RRH1 (14.3 kbp, NC_016651) and *Lactococcus* phage bIL311 (14.5 kbp, NC_002670). Their genome lengths were well within the confidence limit predicted for T = 3 tailed phage capsids, 8.1–17.1 kbp. But their genome lengths also overlapped with the lower limit predicted for T = 4 tailed phage capsids, in the 13.2–24.5 kbp range. It thus cannot be excluded that these two phages adopt T = 4 instead of T = 3 capsids.

Among the gut metagenome-assembled tailed phages, the putative phages OGQL01007720.1 (4.5 kbp) and OMEC01003054.1 (5.4 kbp) were predicted to adopt T = 4/3 ≈ 1.33 icosahedral capsids. The trihexagonal lattice associated with T = 4/3 ≈ 1.33 has been observed among higher T-numbers for tailed phages [19,20]. However, no major capsid protein or portal protein was annotated in those two genomes as well as in five other genomes predicted to adopt larger T-numbers (Appendix A). The absence of a major capsid protein, however, does not imply that the assembled genomes in our study were incomplete. All seven of these phages encoded at least one phage hallmark gene (portal protein or a large terminase subunit) and possessed terminal repeats at their contig termini, which are indicative of a closed genome assembly. One possibility explaining the lack of an identifiable major capsid protein is that the correct open reading frame (ORF) was not predicted [67]. Another interpretation is that there could be overlapping open reading frames, although this is unusual among tailed phages [67]. Alternatively, these contigs could be associated to satellite phages, which often lack the major capsid protein and other structural genes. In this case, it cannot be ruled out that the capsid structure adopted would be bigger than the predicted architecture. This is analogous to *Enterobacteria* phage P4 (NC_001609, genome length 11.6 kbp), which is a satellite phage of P2. Phage P4 does not encode a major capsid protein and forms a T = 4 capsid [66], which is larger than the predicted T = 3 capsid based on its genome length.

Nine metagenome-assembled tailed phages that displayed genomes from 7.3 to 8.5 kbp and were predicted to adopt a T = 7/3 ≈ 2.33 icosahedral capsid (Figure 6b). This group of diverse phages encoded major capsid proteins (Appendix A), providing confidence to this prediction. Nonetheless, the capsid predicted adopted a snub hexagonal lattice, which has not been observed among larger tailed phages [19]. The same applies to the next group of ten tailed phages, which were predicted to adopt the T = 4/3 + 2/$\sqrt{3}$ ≈ 2.49 capsid architecture. This capsid is associated with the rhombitrihexagonal lattice, which has not been observed among tailed phages either. If these predictions are confirmed, these would be the first viruses known to adopt regular snub hexagonal and rhombitrihexagonal lattices [19]. Alternatively, these two groups of phages could form elongated T = 1 capsids or icosahedral T = 3 capsids because the genome length of these phages is close to the lower 95% confidence interval predicted for T = 3 capsids, 8.1 kbp (Figure 4). No tailed phages were predicted to adopt the smallest icosahedral capsid, T = 1.

Additional studies will be necessary to determine if such small-tailed phage capsids exist. However, the bioinformatic identification of capsid architectures among uncultured phages adds to the trend of computational-based methods directing discoveries in phage biology. The assembly of co-occurring contigs in metagenomic samples recently led to the bioinformatic discovery of crAssphage (cross-assembly phage) [68]. CrAssphage is the most common group of phages in humans, but it had not been isolated previously [69]. The bioinformatic discovery fueled a search for this virus, and a crAssphage candidate, ΦCrAss001, was finally isolated after expanding the initial taxonomy of crAssphage [70,71,72]. The genome size of this candidate is 102 kbp. Our model predicts that it should adopt a T = 12 or T = 13 capsid architecture. This genome length and capsid architecture coincide with the frequency peak observed in the distribution of assembled genomes at ~98.2 kbp (Figure 5c). This peak is absent in the distribution of isolated genomes (Figure 5a). This result indicates that the architecture of crAssphage-like viruses has been under-sampled among isolated phages. The peaks associated with crAssphage-like genome lengths as well as small genome size phages are the only regions that differ significantly between the isolated and assembled genomes (Figure 5). This suggests that, besides these exceptional cases, the structural analysis of the current pool of isolated phages is probably a good proxy to account for the diversity of tailed phage architectures in the biosphere.

The allometric models developed in this study apply to tailed phages and other viruses in the *Duplodnaviria* realm because they share the same building block (HK97-fold major capsid protein) and genome packing strategy (dsDNA packed at high densities). The scaling exponents or geometric pre-factors differ for viruses that build their capsids with different capsid protein folds or pack their genome in other chemical and physical configurations [36]. Therefore, it is not strictly possible to validate the genome length and capsid size predictions for T = 1 and T = 3 capsids using viruses that do not belong to the *Duplodnaviria* realm. However, divergences between the different allometric models should reduce for small capsids because, in these capsids, the surface-to-volume ratio is higher. Viruses that pack their genome at lower densities than tailed phages tend to accumulate genome layers near the interior wall of the capsid and leave a void in the interior [73,74]. As capsids become smaller, the volume near the interior capsid wall, which contains the genome, will become larger than the central volume without the genome. Therefore, the model predictions could be estimated qualitatively with existing T = 1 viruses in realms other than *Duplodnaviria*. The model would provide an upper limit in the expected amount of genome stored, while the discrepancies in the capsid size should be associated—in a first approximation—to differences in the typical amino acid sequence length of the associated capsid proteins.

All isolated viruses that have been characterized to form T = 1 capsids—excluding satellite viruses and viral-like particles—store their genome as single-stranded DNA (ssDNA), and they belong to the *Microviridae, Parvoviridae,* and *Circoviridae* viral families in the *Monodnaviria* realm [29,75]. Viruses in these families all form capsids using the single-jellyroll fold [14]. However, the capsid proteins differ in size, and the viruses in each family infect different organisms and adopt different capsid sizes and genome lengths [76]. *Microviridae* viruses infect bacteria, display genomes lengths of 4.4 to 6.1 kb (kilobases), and capsid diameters of 30 nm, which include decoration capsid proteins and a single jellyroll capsid protein of 427 amino acids for the reference virus phiX174 (PDB 2BPA). *Parvoviridae* viruses infect vertebrates and insects, display genome lengths of 4 to 6 kb, and capsid diameters of 18–26 nm, with a single jellyroll capsid protein of 519 amino acids for the reference virus adeno-associated virus (PDB 1LP3). *Circoviridae* viruses infect birds and pigs, display genome lengths of 1.8 to 2.0 kb, and capsid diameters of 17 nm, with a single jellyroll capsid protein of 226 amino acids for the reference virus porcine circovirus 2 (PDB 3R0R). The model in our study predicted that tailed phages adopting T = 1 capsids would display a genome length in the range of 1.2–4.4 kbp and a capsid diameter in the range of 19.5–30.1 nm (Figure 4). Since tailed phages store dsDNA, the predicted range must be doubled for ssDNA viruses, that is, 2.4–8.8 kb. This represents an upper limit range because tailed phages store their genome at high densities throughout the entire interior capsid volume. In other words, 8.8 kb would be the maximum genome length for ssDNA viruses (with 95% confidence). The genome lengths of *Microviridae* and *Parvoviridae* viruses fall within the mid-range predictions of our model. The capsid diameter of *Microviridae* viruses is in the upper range, probably due to the presence of the decoration proteins. The capsid diameter of *Parvoviridae* viruses is in the mid-range of the capsid diameters predicted by the model. The case of *Circoviridae* viruses falls on the lower limit of the predicted genome lengths and capsid diameters. This is probably due to the fact that the capsid protein of tailed phages is typically ~300–350 amino acids—much larger than for *Circoviridae* viruses. In any case, the qualitative comparison between the predictions of the T = 1 capsids of tailed phages and the observed T = 1 viruses is consistent.

The comparison with *Microviridae* viruses is particularly appealing because these viruses infect bacteria and follow an analogous life cycle as tailed phages: translocating the genome through the host cell upon infection, assembling an empty procapsid, packing the genome to form the mature capsid, and lysing the host to release the new mature virions [31,32]. The absence of a tail in *Microviridae* viruses suggest that tailed phages in the *Podoviridae* family might be more likely to form T = 1 capsid architectures or small phage capsids. This is consistent with the fact that the smallest isolated tailed phages predicted to form T = 3 capsids were *Podoviridae*.

It has been observed that the genome length of viruses, bacteria, and archaea are smaller at higher temperatures [77,78,79]. This suggests that T ≤ 3 architectures would be more likely to be present in hot environments, such as hydrothermal vents, which can reach temperatures of ~100 °C or higher. Warm-blooded animals may also be a good source of small-tailed phage capsids. The temperature is lower than in hydrothermal vents but higher than in most other environments. Due to medical and economic reasons, metagenomes from warm-animal sources are more accessible and abundant, increasing the odds of finding small-tailed phage capsids. This is consistent with our finding of small capsids predicted among human gut tailed phages.

The discovery and study of small-tailed phage capsids would help interrogate the origin of the ancient *Duplodnaviria* realm and the origin of the last universal common ancestor for phages [30]. The assumption is that the smallest icosahedral capsids were precursors of tailed phage capsids. Therefore, the molecular information and divergence of small-tailed phages could provide insightful clues on how viruses emerged on Earth. These small-tailed phages could also help uncover a potential evolutionary relationship between tailed phages and cellular compartments like encapsulins [14,16,17].

The discovery and characterization of small-tailed phages would also have important biomedical implications. Their shorter genomes would contain fewer genes. It is expected that most of the genes would be involved in structural functions. The conserved folds of these proteins, such as the major capsid protein and the portal, would help narrow the functions of genes in these small genomes. This would facilitate the annotation and functional characterization of these phages, circumventing one of the main issues of phage therapy nowadays. Even in scenarios where phages have been used successfully as therapeutics, the number of unknown gene functions encoded in those phage genomes was significant and may be involved in increasing bacterial virulence [80,81]. Small-tailed phages would make the application of phage therapy more predictable and easier to regulate.

## 5. Conclusions

Small-tailed phages with icosahedral architectures T ≤ 3 have not yet been observed by high-resolution imaging techniques. Here, we proposed that these structures exist in the environment but are challenging to sample due to low abundances. The modeling and bioinformatic approach applied here predicted small icosahedral capsids among isolated and metagenome-assembled tailed phages. The smallest capsid predicted was a trihexagonal T = 4/3 ≈ 1.33. No candidate was found to form the smallest icosahedral capsid T = 1, but we proposed that *Podoviridae* in high-temperature environments, like warm-blooded animals and hydrothermal vents, would be potential candidates for the missing T = 1 tailed phage capsids. The discovery of these small-tailed phages would be transformative for the study of viral evolution as well as biomedical and biotechnological applications.

## Figures and Tables

**Figure 1 microorganisms-08-01944-f001:**
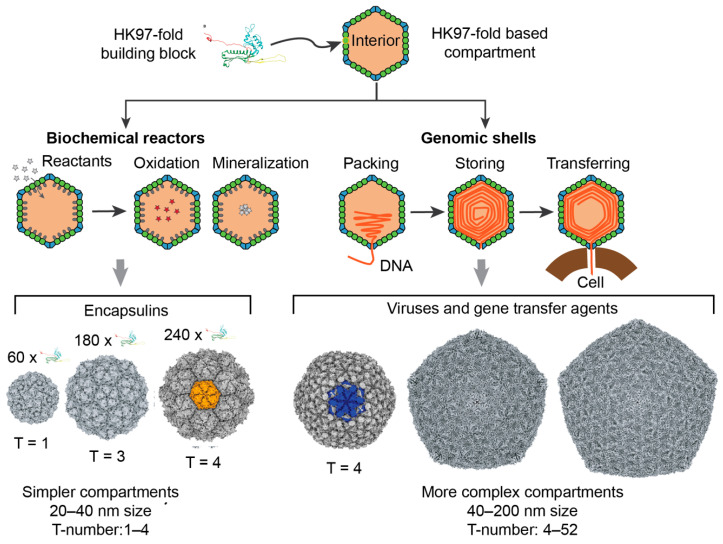
HK97-fold protein compartments. The left side of the figure focuses on encapsulins, nanocompartments responsible for chemical storage and biochemical reactions in bacteria and archaea. The right side of the figure focuses on viruses and gene transfer agents. The viruses belong to the realm of *Duplodnaviria*. Tailed phages in the phylum *Uroviricota* and class *Cauviricetes* are the most diverse and abundant representatives of this group.

**Figure 2 microorganisms-08-01944-f002:**
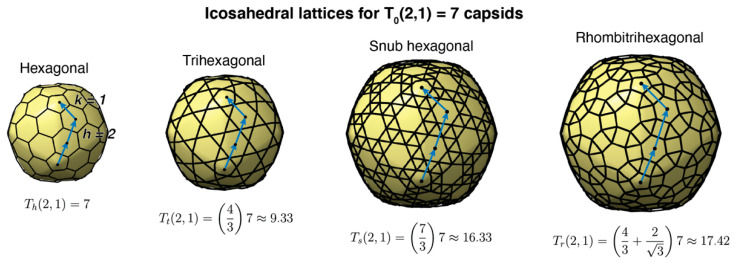
Icosahedral lattices for T(2,1) = 7 capsids. The label on the top displays the name of the lattice. The blue arrows and blue dots display the h and k steps in the hexagonal lattice, h = 2 and k = 1 in this case. The generalized T_h_, T_t_, T_s_, and T_r_ numbers are obtained from the classic T-number multiplied by the lattice factor associated with the minor polygons (triangles and squares) of each lattice: h (hexagonal), t (trihexagonal), s (snub hexagonal), and r (rhombitrihexagonal).

**Figure 3 microorganisms-08-01944-f003:**
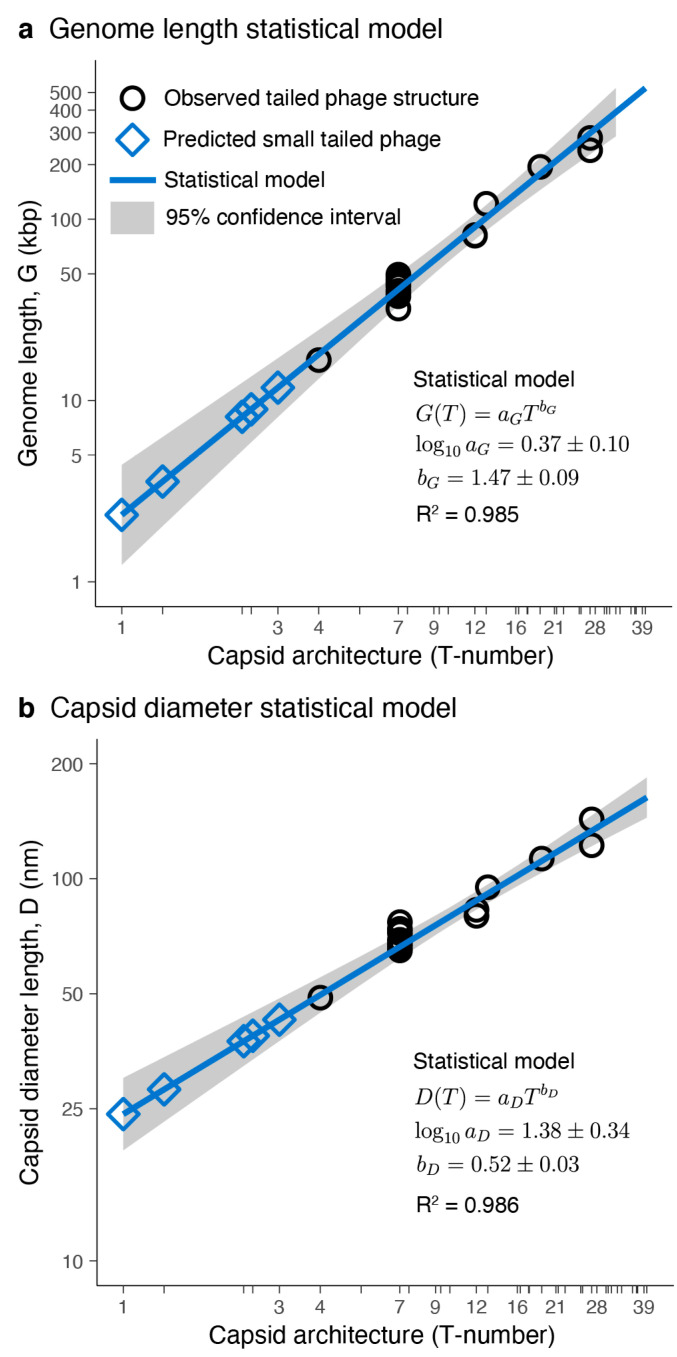
Tailed phage statistical models. Genome length (**a**) and capsid diameter (**b**) plotted as a function of capsid architecture for the studied tailed phage structures (black circles) and the predicted small-tailed phage structures (blue diamonds). (**a**,**b**) The solid blue lines and grey areas correspond, respectively, to the mean values and 95% confidence interval predicted from the statistical model. The mean values and standard errors of the fitted parameters, as well as the coefficient of determination (R^2^), are displayed.

**Figure 4 microorganisms-08-01944-f004:**
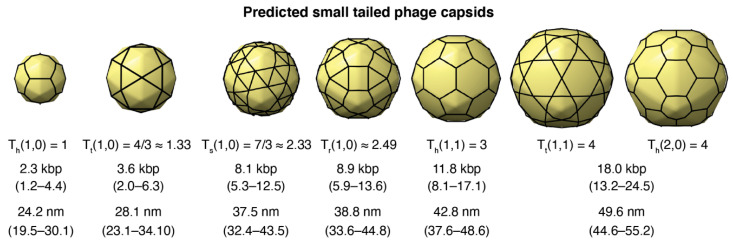
Predicted small-tailed phage capsids. The sequence of three-dimensional (3D) capsid icosahedral models generated with the new hkcage tool in ChimeraX and available in https://github.com/luquelab/hkcage.git. The information below each structure corresponds to the T-number architecture, (h,k) steps, lattice (h: hexagonal, t: trihexagonal, s: snub hexagonal, and r: rhombitrihexagonal), the average predicted genome length (95% confidence interval), and the average predicted capsid diameter (95% confidence interval).

**Figure 5 microorganisms-08-01944-f005:**
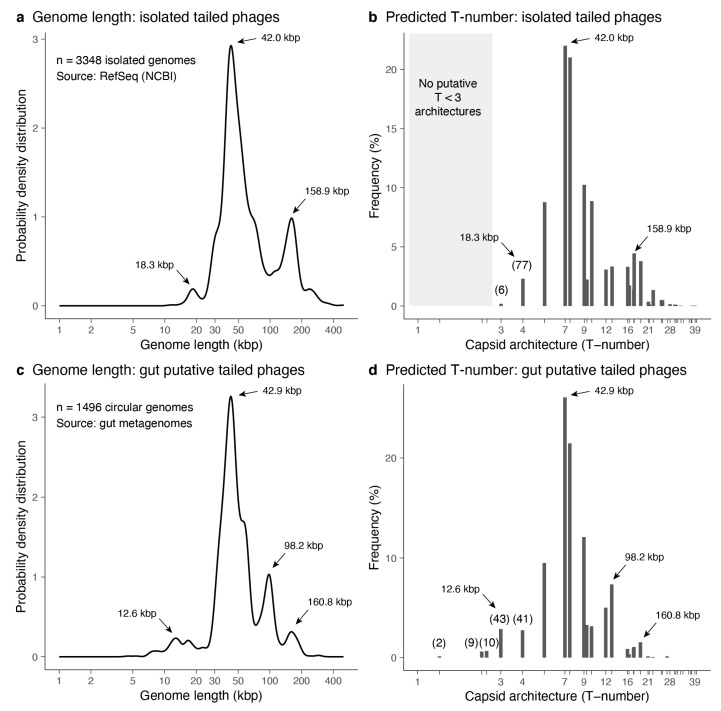
Predicted architectures among tailed phage isolates. (**a**) Genome length distribution for tailed phage isolates obtained from the NCBI Reference Sequence Database. (**b**) Frequency (percentage) of predicted T-number architectures from the genome lengths of isolated tailed phages. The grey area highlights the absence of T < 3 architectures. (**c**,**d**) Genome length distribution and predicted T-number architectures for putative gut tailed phages. (**a**–**d**) The arrows indicate the significant peaks of the genome length distribution and the associated T-number architectures. (**b**,**d**) The parenthesis indicates the number of predicted T ≤ 4 phage capsid architectures.

**Figure 6 microorganisms-08-01944-f006:**
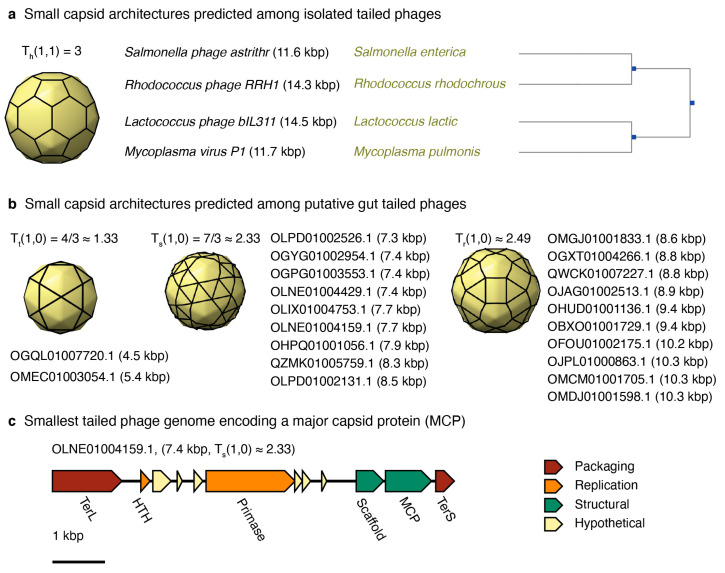
Predicted small-tailed phages. (**a**) List of isolated tailed phages predicted to adopt a T = 3 capsid architecture. The panel displays the phage name, genome length, host, and hosts’ phylogenetic tree. (**b**) List of putative gut tailed phage genomes predicted to adopt T ≤ 3 capsid architectures. The list displays the contig name and genome length in parenthesis. (**c**) Genome map of the smallest genome encoding a major capsid protein (MCP). The scale is 1 kbp. The genes are grouped by function: packaging (red), replication (orange), structural (green), and hypothetical (yellow).

**Table 1 microorganisms-08-01944-t001:** Summary of measured structural properties. * The sphere factor ranged from 0 (polyhedral) to 1 (spherical). ^†^ Maximum (icosahedral) diameter determined from the vertex (5-fold) to vertex (5-fold).

Property	Range	Property	Range
Capsids analyzed	23	Genome size	17–280 kbp
T-number	4–27	Genome density	0.34–0.62 bp/nm^3^
Interior sphericity *	0.25–0.75	Interior surface	5.08 × 10^3^–4.32 × 10^4^ nm^2^
Exterior sphericity *	0.14–0.55	MCP interior area	19–26 nm^2^
Capsid diameter ^†^	49–143 nm	Exterior surface	6.55 × 10^3^–5.19 × 10^4^ nm^2^
Capsid thickness	3–8 nm	MCP exterior area	24–35 nm^2^
Interior volume	3.36 × 10^4^–8.22 × 10^5^ nm^3^	MCP ratio (%)	15–43%

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
