# Peer review of "The Missing Tailed Phages: Prediction of Small Capsid Candidates"

_microorganisms, 2020, doi:10.3390/microorganisms8121944_

Round 1
Reviewer 1 Report
This manuscript describes a modeling approach coupled with bioinformatics to predict possible low T-number tailed phages, which have yet to be observed. The authors use a training set of structurally characterized tailed phages and predict architectures of tailed phages from metagenomics data using the resulting model. The manuscript is accessible and well-written, may prove useful as a guide for experimental confirmation of the small T-number tailed phages. I have a few comments that the authors may wish to address, that will hopefully serve to strengthen the work given the absence of experimental validation at this time:
- The model quantifies what appears to be an intuitive result with respect to the relationship between T number, diameter, and capsid surface. Thus the manuscript could acknowledge this more explicitly (it is touched upon on page 7), or spend a bit more time explaining to a less expert reader how the model outperforms intuition. For example, how does the model reveal information that would be otherwise overlooked with respect to these relationships? Is the connection to genome length all that is brought? If so, would it not be as easy to plot genome length to the cube of diameter, for example? How much does such simplification lead to additional error?
- It would be useful to demonstrate the model’s predictive power by validating one of the predictions, and this is unfortunately not possible as no known tailed phages are of T<3. Could the authors instead apply the approach to non-tailed phage, which do include members such as T=1? By using some data as a training set and others as a validation, it would lend more support to the findings for tailed phage.
- I appreciate the critical review of the predictions, and support the adjustments made to the predictions for the T=3 capsids (ie, that of 6 predicted T=3 tailed phage, four appear likely). Was this same analysis performed for each T number predicted? The manuscript ends rather abruptly after reporting the number of predictions for each lower T number, and I was eager for more information.
Author Response
We have addressed the reviewers’ comments in the revised manuscript, tracking changes. Below we provide a point-by-point response to the reviewers’ comments and indicate the revised manuscript’s lines with changes. Our answers are indented as bullet points and start with the word “Response” in bold text.
Reviewer one
This manuscript describes a modeling approach coupled with bioinformatics to predict possible low T-number tailed phages, which have yet to be observed. The authors use a training set of structurally characterized tailed phages and predict architectures of tailed phages from metagenomics data using the resulting model. The manuscript is accessible and well-written, may prove useful as a guide for experimental confirmation of the small T- number tailed phages. I have a few comments that the authors may wish to address, that will hopefully serve to strengthen the work given the absence of experimental validation at this time:
- Response: We thank the reviewer for carefully evaluating our manuscript. We have replied to the reviewer’s main concerns below and provided the lines associated with the changes in the revised manuscript.
- The model quantifies what appears to be an intuitive result with respect to the relationship between T number, diameter, and capsid surface. Thus, the manuscript could acknowledge this more explicitly (it is touched upon on page 7), or spend a bit more time explaining to a less expert reader how the model outperforms intuition. For example, how does the model reveal information that would be otherwise overlooked with respect to these relationships? Is the connection to genome length all that is brought? If so, would it not be as easy to plot genome length to the cube of diameter, for example? How much does such simplification lead to additional error?
- Response: Our theoretical model derived from biophysical principles provided a justification to the scaling exponents obtained in the statistical analysis. The result is intuitive in the case of tailed phages because the tailed phage genome fills up the interior volume of the capsid. Our statistical approach, however, is not based on this intuitive fact and would lead to a different scaling if the viruses analyzed were not filling the interior capsid volume, as would be the case for RNA icosahedral viruses. As the reviewer points out, in our data, the genome length follows a power function relationship with the cube of the diameter. We could have shown this relationship, but we chose instead to focus on the relationship between genome length and T-number because our goal, rather than proving the apparently intuitive scaling relationship, was to develop a model that could be applied to empirical data to relate genome lengths and capsid architectures. The volume of the capsid is only accessible accurately when a high-resolution structure is available. Therefore, relying on the volume would have precluded a direct connection between the capsid architecture and the phage genomes (isolated and assembled). Using the volume relationship would have added an extra step (genome length ïƒ volume ïƒ T-number), increasing the uncertainty of the prediction. There would be uncertainty in the relationship between genome length and volume as well as volume and T-number. Using the direct relationship between genome length and T-number circumvents this issue. We have added a paragraph clarifying this important insight in the Discussion as requested by the reviewer (lines 330–343). The quotes below provide the excerpt of the text in the revised manuscript.
- “The agreement between the theoretical and statistical scaling exponents for the genome length and the T-number quantified rigorously the expected relationships for tailed phages. Tailed phages pack their genome at a similar density, ~0.5 bp/nm3. This leads to a genome length scaling with the volume of the capsid. Since the T-number is proportional to the surface of the capsid and the exposed surface of the capsid protein is conserved, this led to a scaling of 3/2 (statistically, 1.47±0.09) relating the genome length and the capsid architecture expressed in terms of the T-number. The genome length is accessible through molecular biology methods while the T-number can be obtained from high-resolution microscopy methods. The derived relationship between the genome length and T-number, thus, provided a direct approach to estimate the capsid architectures from the abundant molecular biology data. It also reduced the uncertainty of incorporating intermediate relationships, for example, with the capsid volume. The application of such relationship to isolated and assembled tailed phage genomes provided a method to predict the existence of missing small tailed phage capsids.”
- It would be useful to demonstrate the model’s predictive power by validating one of the predictions, and this is unfortunately not possible as no known tailed phages are of T<3. Could the authors instead apply the approach to non-tailed phage, which do include members such as T=1? By using some data as a training set and others as a validation, it would lend more support to the findings for tailed phage.
- Response: The model derived in our study cannot be applied technically to viruses outside the Duplodnaviria realm because they differ in the capsid protein fold and physicochemical storage of the viral genome. However, it is possible to perform a qualitative comparison for the smallest capsid, T = 1. We have included two paragraphs in the discussion explaining the caveats of this comparison and the case for viruses from three different families. The model predictions are consistent with these T=1 viruses. This has been reflected in lines 412-453. Below we include the excerpts from the revised manuscript between quotes.
- “The allometric models developed in this study apply to tailed phages and other viruses in the Duplodnaviria realm because they share the same building block (HK97-fold major capsid protein) and genome packing strategy (dsDNA packed at high densities). The scaling exponents or geometric prefactors differ for viruses that build their capsids with different capsid protein folds or pack their genome in other chemical and physical configurations [36]. Therefore, it is not strictly possible to validate the genome length and capsid size predictions for T = 1 and T = 3 capsids using viruses that do not belong to the Duplodnaviria However, divergences between the different allometric models should reduce for small capsids because, in these capsids, the surface-to-volume ratio is higher. Viruses that pack their genome at lower densities than tailed phages tend to accumulate genome layers near the interior wall of the capsid and leave a void in the interior [72,73]. As capsids become smaller, the volume near the interior capsid wall, which contains the genome, will become larger than the central volume without the genome. Therefore, the model predictions could be estimated qualitatively with existing T = 1 viruses in realms other than Duplodnaviria. The model would provide an upper limit in the expected amount of genome stored, while the discrepancies in the capsid size should be associated—in a first approximation—to differences in the typical amino acid sequence length of the associated capsid proteins.”
- “All isolated viruses that have been characterized to form T = 1 capsids—excluding satellite viruses and viral-like particles—store their genome as single-stranded DNA (ssDNA); they belong to the Microviridae, Parvoviridae, and Circoviridae viral families in the Monodnaviria realm [29,74]. Viruses in these families all form capsids using the single-jellyroll fold [14]. However, the capsid proteins differ in size, and the viruses in each family infect different organisms and adopt different capsid sizes and genome lengths [75]. Microviridae viruses infect bacteria, display genomes lengths of 4.4 to 6.1 kb (kilobases), and capsid diameters of 30 nm, which include decoration capsid proteins and a single jelly roll capsid protein of 427 amino acids for the reference virus phiX174 (PDB 2BPA). Parvoviridae viruses infect vertebrates and insects, display genome lengths of 4 to 6 kb, and capsid diameters of 18-26 nm with a single jelly roll capsid protein of 519 amino acids for the reference virus adeno-associated virus (PDB 1LP3). Circoviridae viruses infect birds and pigs, display genome lengths of 1.8 to 2.0 kb, and capsid diameters of 17 nm with a single jelly roll capsid protein of 226 amino acids for the reference virus porcine circovirus 2 (PDB 3R0R). The model in our study predicted that tailed phages adopting T = 1 capsids would display a genome length in the range of 1.2–4.4 kbp and a capsid diameter in the range of 19.5–30.1 nm (Figure 4). Since tailed phages store dsDNA, the predicted range must be doubled for ssDNA viruses, that is, 2.4–8.8 kb. This represents an upper limit range because tailed phages store their genome at high densities throughout the entire interior capsid volume. In other words, 8.8 kb would be the maximum genome length for ssDNA viruses (with 95% confidence). The genome lengths of Microviridae and Parvoviridae viruses fall within the mid-range predictions of our model. The capsid diameter of Microviridae viruses is in the upper range, probably due to the presence of the decoration proteins. The capsid diameter of Parvoviridae viruses is in the mid-range of the capsid diameters predicted by the model. The case of Circoviridae viruses falls on the lower limit of the predicted genome lengths and capsid diameters. This is probably due to the fact that the capsid protein of tailed phages is typically ~300-350 amino acids—much larger than for Circoviridae In any case, the qualitative comparison between the predictions of the T = 1 capsids of tailed phages and the observed T = 1 viruses is consistent.”
- I appreciate the critical review of the predictions, and support the adjustments made to the predictions for the T=3 capsids (ie, that of 6 predicted T=3 tailed phage, four appear likely). Was this same analysis performed for each T number predicted? The manuscript ends rather abruptly after reporting the number of predictions for each lower T number, and I was eager for more information.
- Response: The predictions were scrutinized in detail for the phage genomes associated with T ≤ 3 architectures, which were the goal of our study. The phage genomes associated with capsids predicted to adopt larger architectures were not investigated in depth; this was beyond the scope of this article. We have clarified this for isolated genomes in lines 155–157 and for assembled genomes in lines 174–176.
- There is, however, an important result among larger capsids that we overlooked in the initial version of the manuscript. The bioinformatically discovered crAssphage group has genome lengths that are associated with a peak in the distribution of genome assembled phages that is missing in the distribution of isolated phages. Our model predicts, thus, that the pool of T=12–13 architectures has been under-sampled among isolated phages compared to environmental phages. We have clarified this in the Discussion in lines 387–411. We have added the excerpt of text below between quotes. We hope that this insightful observation makes our study more rounded regarding low and large T-numbers as recommended by the reviewer. We have also included data files with the predicted genome length ranges for each T-number (new Data File 4), predicted capsid diameter ranges for each T-number (new Data File 5), predicted T-number for all isolated genomes analyzed (new Data File 7), and predicted T-number for all metagenome-assembled genome analyzed (new Data File 9).
- “Additional studies will be necessary to determine if such small tailed phage capsids exist. However, the bioinformatic identification of capsid architectures among uncultured phages adds to the trend of computational-based methods directing discoveries in phage biology. The assembly of co-occurring contigs in metagenomic samples led recently to the bioinformatic discovery of crAssphage (cross-assembly phage) [67]. crAssphage is the most common group of phages in humans, but it had not been isolated previously [68]. The bioinformatic discovery fueled a search for this virus, and a crAssphage candidate, ΦCrAss001, was finally isolated after expanding the initial taxonomy of crAssphage [69–71]. The genome size of this candidate is 102 kbp. Our model predicts that it should adopt a T = 12 or T = 13 capsid architecture. This genome length and capsid architecture coincide with the frequency peak observed in the distribution of assembled genomes at ~98.2 kbp (Figures 5c). This peak is absent in the distribution of isolated genomes (Figure 5a). This result indicates that the architecture of crAssphage-like viruses has been undersampled among isolated phages. The peaks associated with crAssphage-like genome lengths as well as small genome size phages are the only regions that differ significantly between the isolated and assembled genomes (Figures 5). This suggests that, besides these exceptional cases, the structural analysis of the current pool of isolated phages is probably a good proxy to account for the diversity of tailed phage architectures in the biosphere.”
Reviewer 2 Report
Review of Manuscript “The Missing Tailed Phages: Prediction of Small Capsid Candidates“ by Luque et al..
Tailed phages with icosahedral structure architectures in the range of T=1 to T=3 have not been identified yet.
Based on the analysis of the structural capsid features of 23 phages with a genome size ranging from 17 to 280 kbp, the authors present a statistical model to relate the genome size and the capsid diameter to the capsid architecture (T number). Within a 95% confidence interval, the genome sizes of putative small tailed phage capsids were predicted to 2.34 kbp (T=1) to 11.78 kbp (T=3).
Based on these findings, 4 tailed phages from a databank of isolated phage genomes were predicted to form a T=3 architecture, whereas twenty-one metagenome-assembled tailed phages were predicted to form T≤3 architectures. Since many of the sequences based on assembled metagenomes did not contain a putative ORF for a major capsid protein, these are probably predominantly only partial sequences.
However, the detailed analysis strongly suggests that at least phages with a T=3 architecture exist.
The manuscript is well written and the date is very clearly presented. Especially the introduction section is very informative,
Therefore, if the minor points raised below were addressed in a revised version of the manuscript, I would strongly recommend publication.
Minor points:
1) Fig. 5: In fig. 5a and 5b the most dominant peak in genome is depicted at 41.0 kb, whereas in the text (lines 255 and 261, respectively) a value of 42.0 kb is given.
2) The columns in the data files 3 to 5 could be labeled in more detail.
Author Response
We have addressed the reviewers’ comments in the revised manuscript, tracking changes. Below we provide a point-by-point response to the reviewers’ comments and indicate the revised manuscript’s lines with changes. Our answers are indented as bullet points and start with the word “Response” in bold text.
Reviewer two
Tailed phages with icosahedral structure architectures in the range of T=1 to T=3 have not been identified yet.
Based on the analysis of the structural capsid features of 23 phages with a genome size ranging from 17 to 280 kbp, the authors present a statistical model to relate the genome size and the capsid diameter to the capsid architecture (T number). Within a 95% confidence interval, the genome sizes of putative small tailed phage capsids were predicted to 2.34 kbp (T=1) to 11.78 kbp (T=3).
Based on these findings, 4 tailed phages from a databank of isolated phage genomes were predicted to form a T=3 architecture, whereas twenty-one metagenome-assembled tailed phages were predicted to form T≤3 architectures. Since many of the sequences based on assembled metagenomes did not contain a putative ORF for a major capsid protein, these are probably predominantly only partial sequences.
- Response: We thank the reviewer for carefully evaluating our manuscript.
- Regarding the observation “assembled metagenomes did not contain a putative ORF for a major capsid protein, these are probably predominantly only partial sequences”, we would like to highlight that 14 of the 21 metagenome-assembled tailed phages with a predicted T ≤3 architecture encoded a major capsid protein (Figure S3). The absence of a major capsid protein, however, does not necessarily reflect that the assembled genomes in our study were incomplete. All of these phages encoded at least one phage hallmark gene (portal protein or a large terminase subunit) and possessed terminal repeats at their contig termini, indicative of a closed genome assembly. For the 7 contigs that lacked a major capsid protein, one possibility is that the open reading frame for the major capsid protein gene was not identified [66]. Another interpretation is that there could be overlapping open reading frames, although this is unusual among tailed phages [66]. Alternatively, they could be associated to satellite phages, which often lack the major capsid protein and other structural genes. In this case, it cannot be ruled out that the capsid structure adopted would be bigger than the predicted architecture. This is analogous to Enterobacteria phage P4 (NC_001609, genome length 11.6 kbp), which is a satellite phage of P2. Phage P4 does not encode a major capsid protein and forms a T = 4 capsid [65], which is larger than the predicted T = 3 capsid based on its genome length. This insight clarifying the comment from the reviewer has been added to the revised manuscript in lines 164–172 and lines 359–374. Below we provide the specific excerpt of text between quotes.
- “Only ‘circular’ contigs with relatively long direct repeats (50 – 200 bp) at their termini were searched for open reading frames (ORFs). The contigs were also required to include a known phage marker profile, that is, the terminase large subunit, major capsid protein, or portal protein.”
- “The absence of a major capsid protein, however, does not necessarily reflect that the assembled genomes in our study were incomplete All of these phages encoded at least one phage hallmark gene (portal protein or a large terminase subunit) and possessed terminal repeats at their contig termini, indicative of a closed genome assembly. For the 7 contigs that lacked a major capsid protein, one possibility is that the open reading frame for the major capsid protein gene was not identified [65]. Another interpretation is that there could be overlapping open reading frames, although this is unusual among tailed phages [65]. Alternatively, they could be associated to satellite phages, which often lack the major capsid protein and other structural genes. In this case, it cannot be ruled out that the capsid structure adopted would be bigger than the predicted architecture. This is analogous to Enterobacteria phage P4 (NC_001609, genome length 11.6 kbp), which is a satellite phage of P2. Phage P4 does not encode a major capsid protein and forms a T = 4 capsid [64] , which is larger than the predicted T = 3 capsid based on its genome length.”
However, the detailed analysis strongly suggests that at least phages with a T=3 architecture exist.
The manuscript is well written and the date is very clearly presented. Especially the introduction section is very informative.
Therefore, if the minor points raised below were addressed in a revised version of the manuscript, I would strongly recommend publication.
- Response: We are glad that the reviewer found the manuscript “well-written” and “clearly presented,” that the introduction was “very informative,” and that “strongly recommend publication.” Below we have addressed the minor points raised by the reviewer.
Minor points:
1) Fig. 5: In fig. 5a and 5b the most dominant peak in genome is depicted at 41.0 kb, whereas in the text (lines 255 and 261, respectively) a value of 42.0 kb is given.
- Response: The correct value was 42.0 kb. We have edited the label in Figures 5a and 5b to reflect this change.
2) The columns in the data files 3 to 5 could be labeled in more detail.
Response: We used more descriptive labels in those files. In the revised files, the first column corresponds to the identification number. The second column has the genome length in base pairs. The labels used are “contig_id” and “assembled_genome_length_bp” in Data Files 3 (new Data File 3) and 5 (new Data File 7), and “NCBI_id” and “genome_length_bp” in Data File 4 (new Data File 6).